# Practical recommendations from a multi-perspective needs and challenges assessment of citizen science games

**Joshua Aaron Miller**[1]*, **Libuše Hannah Vepřek**[2], **Sebastian Deterding**[3], **Seth Cooper**[1]

**1** Northeastern University, Boston, Massachusetts, United States of America, **2** Ludwig Maximilian University of Munich, Munich, Germany, **3** Imperial College London, London, United Kingdom

* miller.josh@northeastern.edu

## Abstract

Citizen science games are an increasingly popular form of citizen science, in which volunteer participants engage in scientific research while playing a game. Their success depends on a diverse set of stakeholders working together–scientists, volunteers, and game developers. Yet the potential needs of these stakeholder groups and their possible tensions are poorly understood. To identify these needs and possible tensions, we conducted a qualitative data analysis of two years of ethnographic research and 57 interviews with stakeholders from 10 citizen science games, following a combination of grounded theory and reflexive thematic analysis. We identify individual stakeholder needs as well as important barriers to citizen science game success. These include the ambiguous allocation of developer roles, limited resources and funding dependencies, the need for a citizen science game community, and science–game tensions. We derive recommendations for addressing these barriers.

## 1 Introduction

Citizen **science** (also called community science [1] or participatory science [2], see Eitzel et al. [3]) is a growing field and model of science wherein volunteers of the general public assist scientific research to produce scientific knowledge [1, 4–7]. There are many modes and forms of involvement, ranging from bottom-up—where citizen scientists themselves initiate the project [8, 9]—to top-down approaches initiated by professional scientists and research teams. In most cases, volunteers engage with scientific research projects by collecting or analyzing research data [10]. For example, participants can fill in surveys (e.g., the *Dutch flu-tracker* [7]), collect and upload data to a platform (e.g, iNaturalist, https://www.inaturalist.org/) or play games. In this paper, we focus on the latter, on citizen science games (CSGs), sometimes called Games With a Purpose [GWAPS; 11]. CSGs arose with the rise of gamification in the late 2000s and were seen as a perspective-shifting technology that redefined online citizen science engagement [12, 13]. Now, CSGs are being used both for education and research, solving real world problems and inspiring the next generation of scientists [14–17].

Citizen science has proven to be a useful means of scientific knowledge production [e.g., 18–20], extending the capabilities of professional scientists with the power of crowdsourcing.

**Data Availability Statement:** All relevant data are within the paper.

**Funding:** L.H.V.'s research was funded by the Deutsche Forschungsgemeinschaft (DFG –

German Research Foundation; https://www.dfg.de/ ) – 464513114. The funders had no role in study design, data collection and analysis, decision to publish, or preparation of the manuscript.

**Competing interests:** The authors have declared that no competing interests exist.

However, its mode of production has also raised concerns and critiques regarding, e.g., the exploitation of citizen scientist labor, data ownership, data sharing, conflicts of interest [21–25], or inclusion and diversity [25–27].

CSGs are complex socio-technical "system assemblages" [28] that span many non-human as well as different human actors or stakeholders groups, including, e.g., volunteers/players, scientists, and game designers and developers. Among these, much research has examined exactly how CSGs engage their *player audiences* as a primary stakeholder group [29–32], since CSGs aim to attract a wider audience of citizen scientists through the motivational power of games. Such studies also identify challenges players describe in engaging with CSGs, including a steep learning curve and lack of understanding [33], mixed feelings on gamification [34], a lack of fun, engaging play [35], poor user interfaces [36] and support for "dabbling" [37], as well as a need for building community and providing good scientific feedback and communication about the project [32, 38].

In contrast to the substantive player literature, there does not exist much literature on the *scientist*'s perspective [39, 40]. Thus, Golumbic et al. analyzed found that participating scientists' motives and views were often less public-minded than the wider discourse around citizen science. Another study on the "OPAL" (Open Air Laboratories) project in England [23] suggested that scientists were concerned about ethical dimensions on the use of public data and data quality. There are even fewer studies on other stakeholders of citizen science projects, such as policymakers, academics, resource managers, governments, the private sector, or residents of affected environmental areas [12, 41, 42]. One perspective noticeably absent is that of the development teams of CSGs.

We consider their perspectives critical because they already comprise an interdisciplinary breadth of and possible tensions between stakeholders: the project leads, scientists, software developers, game designers, and community managers. Creating a CSG requires expertise in science, game design, software engineering, marketing, communications, and more. Understanding how CSGs operate—from the perspectives of players, researchers, developers, educators, students, and everyone else involved in these projects—can not only contribute to improving models of public participation in scientific research, but also to our understanding of interdisciplinary teams. Moreover, each discipline—each stakeholder group—has unique needs with respect to how they can most effectively contribute to CSG operations.

In this article, we therefore analyze the needs and challenges of the individual stakeholder groups of CSGs, which have so far mainly been considered separately or not at all, in order to understand the recurring barriers in existing case studies and active citizen science games and provide recommendations on how challenges can be overcome and needs can be satisfied for all stakeholder groups.

This article thus also helps address the need highlighted in prior literature [23] not to overlook the problems that can arise in citizen science projects, especially given the great enthusiasm found in today's citizen science literature.

Our needs and challenges assessment was driven by the following research questions:

RQ1. What are the needs of each stakeholder group involved in CSGs?

RQ2. What challenges does each stakeholder group face? What barriers exist which prevent or hinder their contribution?

RQ3. What overarching factors can be derived from the individual perspectives that most strongly shape and influence the development and maintenance of—and participation in—CSGs?

To answer these questions, we conducted a qualitative study spanning two years of ethnographic research and 57 interviews involving stakeholders from ten CSGs: *ARTigo*, *Eterna*, *Eyewire*, *Foldit*, *Forgotten Island*, *Happy Match*, *Reverse the Odds*, *Quantum Moves 2*, *Skill Lab*: *Science Detective*, and *Stall Catchers*. Using a combination of grounded theory and reflexive thematic analysis, we produce descriptive summaries of the needs and challenges identified for each stakeholder group as well as narrative themes which represent issues involving multiple stakeholder groups. Our Discussion summarizes the convergence of these themes and provides recommendations—both from our research and directly from our participants—on how challenges can be overcome and needs can be satisfied for all stakeholder groups. Recommendations include researching previous lessons learned in CSG development (and deciding whether a CSG is right for your project) before creating a new CSG; assigning clear roles to development team members; designing to facilitate knowledge transfer; focusing on community building; and focusing on the entertainment (and gamification) aspects. See Table 4 for a summary of recommendations.

## 2 Materials and methods

The present work is an interdisciplinary collaboration, merging a human-computer interaction study conducted by the first author and an ethnographic field study by the second author. During conversation, we realized that there was a need and benefit in combining our work to analyze stakeholder perspectives. Therefore, we divide this section into the independent methodologies used, followed by a description of the present analysis which combines both datasets. Note that because these studies were conducted independently prior to a joint analysis, it is possible that some authors are also participants or that some participants were in both studies.

### 2.1 Human-computer interaction study

Purposive sampling was used to recruit researchers, educators, and developers (n = 15) involved with a variety of CSGs. Of the participants involved, 6 self-identified as researchers/scientists, 6 as educators, 7 as developers (including game designers and community managers), and 3 as CSG players themselves (self-identifications were not mutually exclusive across roles). Across these participants, the CSGs and citizen science platforms discussed include: *Eterna*, *Eyewire*, *Foldit*, *Forgotten Island*, *Happy Match*, *Reverse the Odds*, *Quantum Moves 2*, and *Skill Lab*: *Science Detective*. In order to protect the anonymity of participants, no demographic data were collected.

Invitations to participate were sent directly via email and linked to a sign-up form for informed consent and interview scheduling. Each interview began with another verbal check of informed consent followed by an hour (on average) of semi-structured interviewing regarding the participant's involvement with CSGs and their experiences, needs, challenges, and advice pertaining to using CSGs. These questions were tailored to the roles they self-identified with; for example, educators were asked "What challenges do your students face when learning how to use [citizen science game]?" while developers were asked "What challenges does your team face in developing [citizen science game]?" Participants were offered a $15 Amazon gift card as remuneration. The interviews were audio-recorded and then transcribed for further analysis. In total, 14.2 hours (per participant: M = 56.84 minutes; SD = 31.45) of data were collected. Protocols were approved by Northeastern University's Institutional Review Board (NEU #19-11-04).

Prior to the joint analysis, data were analyzed by the first author using reflexive thematic analysis [43, 44] with an orientation toward the ways in which each role conceptualizes and frames their relation to the game and CSG community. The goal of this analysis was to try to understand how each role fits into the larger network of CSG praxis through themes based on lived

experiences. The analysis approach was primarily deductive, latent, and constructionist. Although no single theory grounded the analysis, it was informed by existing literature on citizen science participation [including, but not limited to, 10, 28–30, 32–34, 37, 45, 46] and Gee's Discourse theory [47] as a framework for conceptualizing Discourse around an artifact (i.e., the CSG).

The preliminary analysis occurred in four rounds of iteratively passing through the data to apply codes, merge codes into themes, and return to the data to validate and refine themes. After four rounds, the preliminary themes included: the vague but valuable CSG niche, unclear tutorials, mismatched expectations, game evolution over years, and several other work-in-progress themes. Yet, when the analyst was producing *topics* more than *themes* (see [44]), he sought to collaborate with other authors to help unpack the meanings within the data and find the key narratives worth detailing. This led to the joint analysis described below.

## 2.2 Ethnographic study

The empirical data is based on ethnographic research of three different CSGs, *Stall Catchers*, *ARTigo* and *Foldit*, conducted over two years from the beginning of 2020 until the end of 2021. Data collection followed a cultural anthropological inductive and constructivist grounded theory approach [48].

In the case of *Stall Catchers*, which served as the central example, a "co-laborative" approach (a kind of "joint, epistemic work" [49]) was applied including different ethnographic methods such as participant observation was conducted at the *Human Computation Institute* where the game is developed as well as on the platform itself. Semi-structured interviews were also completed with a range of stakeholders including developers and game participants. This data was supplemented with analysis of the game's computer code and media content such as forum and blog posts, as well as in-game text chat produced by *Stall Catchers* participants. The games *ARTigo* and *Foldit* served as comparative examples. Data collection for these two additional games similarly include ethnographic methods, specifically participant observation on the platforms and further semi-structured interviews.

This article draws particularly on interviews from all three case studies. For interview recruitment of CSG team members, purposive sampling was used. In total, across the three games, 7 developers, 4 project leads, 2 community managers, and 8 scientists were interviewed following a semi-structured interview approach. CSG Participants were invited to join the research project via a collective email or a notice in the CSG forum. In total, 30 semi-structured interviews with participants from the three case studies as well as 12 written interviews were conducted. Because of the open call to participate which required interested participants to send an email or message to Libuše, they do not necessarily represent the overall player base of each CSG but instead may be those who are particularly engaged and/or who were motivated to share their perspective. Interviews lasted between 21 minutes and 2 hours and 16 minutes with an average of around 1 hour. All interviews were audio recorded and transcribed with the consent of participants. Interviews were conducted in English, German, and Dutch and on-English interviews were translated by Libuše.

Building on grounded theory methodology, first introduced by Barney Glaser and Anselm Strauss [50], data collection and analysis were conducted in alternating phases, where outcomes from coding and analysis were used to evaluate and refine continued data collection. All empirical material was analyzed using MAXQDA software for qualitative data analysis.

## 2.3 Joint analysis

Although each study in isolation produced some insights, the authors noticed that a clearer understanding could be produced by re-analyzing the combined data together. In combining their data, the first two authors discussed the topics and themes they had generated so far and looked for similarities and differences across the preliminary results. This dialogue followed a

deductive and constructionist "Big Q" [51] qualitative approach to understand the shared phenomena which descriptively summarize the lived experiences of our stakeholder participants. This included, for example, identifying similarities in described experiences and noting contrasts between how participants describe their intentions versus the media artifacts on their project websites and game systems. Particular attention was given to understanding—theoretically and practically—the lived experiences and interpersonal dynamics of stakeholder groups whose voices are yet unheard in the literature, including third-party researchers collaborating with the first-party scientists involved in CSG development, software developers, game designers, community managers, educators, and students.

We oriented our joint analysis to the research question "What are the needs and challenges of each stakeholder perspective?" As such, our results are divided into two sections: first, we discuss each stakeholder group in isolation, identifying their characteristics and needs. Then, we thematize these findings, highlighting challenges and topics of interest that arose across multiple stakeholder groups and pose issues for CSGs more broadly.

### 2.4 Games studied

*Foldit* (https://fold.it/) is a 3D puzzle game about folding proteins into a better shape through manual manipulation, algorithmic tools, and programmable scripts [18]. *Eterna* (https://eternagame.org/) has similar gameplay but is a 2D puzzle game for RNA instead of proteins [52]. Both *Foldit* and *Eterna* are thus biochemistry games with complex tools and focused on designing novel solutions to problems, rather than solving puzzles in a specific possibility space. *Eyewire* (https://eyewire.org/) is a 3D puzzle game about coloring in a 3D model of a neuron based on data from 2D slices of serial electron microscopy images [20]. *Eyewire* is a neuroimaging task made into a relaxing coloring game with a prominent leaderboard and occasional challenge events. The *Citizen Sort* project includes *Happy Match* (https://citizensort.org/web.php/happymatch), a game about taxonomically classifying moths, sharks, and rays, and *Forgotten Island* (https://citizensort.org/web.php/forgottenisland), a point-and-click adventure game with a minigame for labeling images of moths [53, 54]. *Forgotten Island* is notable for its narrative shown through comic panels and dialogue with a robot non-player character. *Reverse the Odds* (no longer available, cf. https://www.cancerresearchuk.org/get-involved/citizen-science) was a mobile puzzle game based on the classic game *Reversi* with citizen science components for labeling images of cancer slides [55]. *Quantum Moves 2* (https://www.scienceathome.org/games/quantum-moves-2/) is a puzzle game about solving quantum transfer problems using ludic representations of particles and wave function densities [56]. *Skill Lab*: *Science Detective* (https://www.scienceathome.org/games/skill-lab-science-detective/) is a citizen psych-science game [57] about assessing cognitive abilities at population-scale through a variety of mini-games [58]. In the online game *Stall Catchers* (https://stallcatchers.com/main), participants annotate research data presented as short movie sequences in a binary way to help biomedical researchers with Alzheimer's disease research. *ARTigo* (https://www.artigo.org/) is an art history game with the aim of designing a semantic search engine for artworks, in which participants are invited to create keywords and tags for artworks presented in different mini games [59]. To protect all interviewees who generously took the time to talk to us about their experiences and perspectives, all projects are anonymized in our results. See Table 1 below for a summary.

## 3 Results

We identified the following main stakeholder groups: participants, project designers and leads, professionally trained scientists, (software) developers, game designers, community managers, educators, and students. Other stakeholder groups such as funders and journalists play

**Table 1. List of games studied sorted by preliminary analysis.** All games were studied in the subsequent joint analysis.

| HCI Study | Both Studies | Ethnographic Study |
|---|---|---|
| *Eterna*<br>*Eyewire*<br>*Happy Match*<br>*Forgotten Island*<br>*Reverse the Odds*<br>*Quantum Moves 2*<br>*Skill Lab*: *Science Detective* | *Foldit* | *ARTigo*<br>*Stall Catchers* |

important roles for CGSs too, but these groups were considered out of scope for the present analysis since they do not interact directly with the production or consumption of the player experience. The definition and division of the individual stakeholder groups has to a certain extent remained an analytical one as the boundaries between, for example, game designers and developers or project leads and the professionally trained scientists are often blurred in CSGs. However, given that each stakeholder group is associated with different responsibilities and tasks in CSGs, the distinction is fruitful for the analysis to gain insights into the specific needs and challenges associated with certain roles.

In the following section, we present the results of the analysis of the individual stakeholder groups. For each group, we first present a description of the role, then discuss their needs and challenges. See Table 2 for a summary of our analysis.

We will refer to statements from our research participants by an identification number and their role(s) as [C]ommunity manager, [D]eveloper, [E]ducation, [G]ame designer, project [L]ead, [P]articipant, and/or [S]cientist. Some quotes are abridged for readability.

## 3.1 Participants

The participant's role is at the core of every CSG project. Without their voluntary engagement, the purpose of the game could not be met, the team's efforts would be in vain. As one of the

**Table 2. Summary of stakeholder groups and our analysis of their individual needs and challenges.**

| Stakeholder | Description | Needs and Challenges |
|---|---|---|
| [P]articipants | Volunteers motivated by science and entertainment | Fear of submitting bad data; need to have contributions celebrated; need communication with developers and scientists; need for ethical clarity of their role in science; need for all voices to be heard (especially newcomers) |
| Project [L]eads | Managers, usually oversee funding and collaborations | Different communication styles within international teams; difficulty building and maintaining community; challenge of handling all of the different roles |
| Professional [S]cientists | Trained researchers | Challenge of public-facing communication; additional, unusual responsibilities beyond typical scientist duties; discrepancies in funding models between science and game; different goals between scientists and players; difficulties collaborating from outside the core team |
| Software [D]evelopers | Software engineers, usually students or part-time employees | High churn rate; limited onboarding; required to fill multiple roles; accumulative tech debt; volunteerism; bottom-up development; scope creep |
| [G]ame Designers | Sub-role of developers (not their own position) | See overarching theme: Science–Game Tensions |
| [C]ommunity Managers | Liaisons between team and participants; often a sub-role | Dealing with player pushback; mediating difference between player needs and developer needs; dealing with inappropriate player behavior; labor and skillset not valued by team |
| [E]ducators / Students | Users of CSGs in the classroom | Need to understand the game better (concepts, controls, gameplay loop, contribution model, etc.); student hesitancy to experiment; challenges with technology; educators need control over content, more educational tutorials, and better support for tracking student progress |
| Other | External funders, commercial companies in the scientific domain, politicians and policymakers | Not included in this article—recommended as future work |

team members describes: "They're everything. They're the most important part [. . .] of the project" [C16].

Although there exist various ways of engagement of non-professionally trained scientists into scientific research—ranging from self-initiated citizen science projects to those that are designed and implemented by professionally trained researchers and developers—the CSGs informing this paper are all examples of the latter. This must be considered when discussing the characteristics and challenges of the participants stakeholder group as they may vary from other forms of engagement in scientific knowledge production.

In these projects, participants voluntarily decide to contribute to a specific CSG (the exceptional case of school children and students will be discussed in Section 3.5) and actively participate by playing the games at their leisure and thereby contributing to scientific research. In some cases, participants also contribute to CS(G)s in the context of special events at their workplaces, but in this paper we do not further consider this case separately.

The motivations of participants to contribute to citizen science projects in general, according to Land-Zandstra et al., can range from contributing to "real scientific research or to an important cause such as the environment or health" [7], a general interest in the project's research topic, fun, the opportunity to learn something about a specific research field, and social reasons, e.g., to get in contact with people with the same interests (ibid.). In CSGs, additional motivations can be enjoyment and complex challenges of the game [32]. Engaged participation with a CSG often requires both the motivation of contributing to the science of the game and the entertainment value of the game itself [60]. In some cases, participation in CSGs can also be a way of coping with everyday life when this is, for example, marked by an incurable disease like Alzheimer's disease or by a pandemic like the COVID-19 pandemic [61].

**3.1.1 Participant needs and challenges.** Participants face different challenges that mainly arise from the entanglement of science and play in CSGs. In our discussions with participants, some of them expressed concern about submitting bad data that could harm the research. This fear often derives from a lack of knowledge about how individual contributions and the results are calculated. Although out of scope for this article, we refer to previous literature which has investigated how to match participant skills to appropriate tasks and how to celebrate individual contributions [10, 26, 31, 46, 62, 63].

Participants are very focused on their contributions. The moments described as most frustrating for participants often refer to the feeling that their contribution is not valuable. For example, one participant explains that "[i]f you get [bad quality data] too many times, you lose interest because it's, you just fear that your work is meaningless" [P17]. This meaninglessness occurs when the research data analyzed is of bad quality or because technical problems and bugs in the code make it difficult for the participant to contribute in a satisfying and meaningful way.

This challenge is also connected to the next one, which regards communication with the CSG team. Although participants positively mention the possibility to communicate with the team via in-game chats and forums, they express their dissatisfaction with the way developers or the team in general handle bugs and issues reported by players. When asked if they would report bugs to the team, one participant described reporting a bug and seeing the bug still present six months later without a response from the developers. Participants have different understandings of the priorities than the development team:

> "[W]hat I find disappointing is that they are now busy with [some new feature] while they also have bugs which are serious and which actually should have been solved first. [. . .] [M]y focus with my software developer background would be: fix the bugs first before introducing new features." [P18]

This communication problem is likely bound to the lack of information about—and intransparency of—processes within the team and the lack of resources that constrain the developer's work. At the same time, this illustrates the power hierarchies within these kinds of citizen science projects: it is the scientists and developers of CSGs who primarily set research goals and priorities, not the participants.

Lastly, there are two challenges with the role of participants, generally. When developers try to survey the participant community to understand how to better serve them, only the most active players engage or have opinions [C9]. Similar effects have been found with commercial games [64]—ultimately, the voices of new participants need to be heard, but measuring their opinions is challenging.

The second issue with participants, generally, is that they often fall through regulatory cracks due to their ambiguously defined role which relates to our larger finding about ambiguous roles, discussed in Section 4.1, but we focus here on how it affects participants in particular. This becomes apparent in Institutional Review Board (IRB) processes evaluating CSG projects where the role of participants moves between the categories of "human subjects", "research participants" and "scientists" [65–67]. In citizen science in general, and therefore also in CSGs, participants can sometimes even be both researchers and human subjects [66]. It is not uncommon in CSGs that some very engaged participants take over additional roles, such as community management, besides their contribution as participants. As moderators, these individuals take on important tasks to maintain the project—for example, they are often the ones who report bugs or problems, but they also act as intermediaries between the team and the participants. While participants volunteer to take over these roles and are publicly recognized for their additional commitment on the project's website or communication platforms, this nevertheless raises questions about the lines drawn between compensated and uncompensated work. Moreover, these different role understandings not only challenge oversight committees—which so far have been particularly focusing on the protection of human subjects (ibid.)—but also the CSG stakeholders investigated in this paper.

Ultimately, the role of CSG participants in the ethics of scientific research remains a necessary conversation with the greater CSG community.

## 3.2 Project leads

The second stakeholder group is the project leads (or managers / project designers) of CSGs. As project leads often take on many different roles, this stakeholder group is not always distinguishable from other stakeholder groups like scientists or developers, and the lines between the groups can be blurred. We define a project lead to be the role of managing the design, development, and maintenance of the project and the team. Often, the project lead oversees the funding of the project, and when the project lead is also a scientist (which is often the case), they determine the direction(s) of the game's research. Project leads are in charge of the overall direction of the project and formally representing the project, but also *go [. . .] out for collaborations and connecting with the community* (paraphrased field note from [S8]). In many CSG teams, these decisions are jointly discussed among the team and tasks are divided between different team members.

**3.2.1 Project lead needs and challenges.** In total, we identified the following three recurring challenges described by project leads of citizen science games: 1) different communication styles within international teams, 2) difficulty to build and maintain community, 3) challenge to handle all of the different roles. A fourth outstanding challenge is a lack of resources, which weaves into all other challenges. Because of their huge responsibility for the CSGs, project leads have to deal heavily with acquiring and managing resources. However, as this challenge affects almost all stakeholders involved in CSGs, we will discuss resource issues in Section 4.2.

The first challenge we identified for the project lead stakeholder group refers to the distributed team structures and mainly remote collaboration. It is not unusual for CSGs to be developed by a team that consists of team members spread around the country or even the world and from different institutions. While these structures may offer some physical flexibility to one's work, they more often present challenges due to different communication styles and availability within the teams.

Not knowing the target audience and the motivations or needs of the users also makes it incredibly hard to build and maintain a community of a CSG [S8], which forms the second challenge for project leads [ELS15]. "Community" here refers to the collection of CSG participants (or players). "Focus on the community, That's what will make or break your project", says [DP13]. Building a community is of utmost importance for CSGs, as these projects are dependent on the ongoing contributions by participants and being part of a community has been described as motivating by participants [29, 30, 32].

At the same time, building a community is no easy endeavor. There does not exist a generic "how to" approach and every community is unique in its shared ideas and what binds them together. It is even questionable if a community can be formed from the outside or if it has to grow from inside.

The third recurring challenge we identified for project leads is the challenge to handle all different roles. As described in the beginning of this section, the role of project leads is not always clearly defined. Because of the mostly small team sizes and lack of resources (with respect to time, funding, and team members), project leads have to step into all of the different roles and tasks. It is rare that one person can adequately replicate the expertise of many roles, which results in one or more of these jobs being insufficiently performed.

## 3.3 Professional scientists

By "professional scientists," we refer to the professionally-trained researchers who lead the scientific investigation behind a specific CSG. In the case studies examined, they are the ones who define the research questions, the methods, and analysis, and who decide how to include the crowd into conducting the research. Professional scientists also write-up and publish the research results and are involved in funding for the projects. In many but not all CSGs, professional scientists also take over the role as project leads. In this section, we focus on the aspects which especially concern scientific and research tasks within CSGs.

For the interviewed scientists, the purpose of CSGs is both to help science, e.g., by accelerating the analysis of research data, and to connect people to science. Some also stressed the potential of the CSG to educate people, like one scientist who explained this to be the main "mission" of the CSG: "I think that the scientific aspects of it are quite valuable but to me that is definitely secondary in my evaluation of it" [S20].

At the same time, CSGs can help scientists reflect on how to explain and present science to the public. By conducting research in view of the public throughout the process, the scientists get both practice explaining the research as well as feedback on what players understood and how the CSG is effective at assisting this research or not: "Having players involved in development is really, really good. . . and immensely valuable for the research" [DS5].

**3.3.1 Scientist needs and challenges.** In practice, though, scientists face difficulties with the kind of public communications described above. For example, because scientists are so interested in the topic, they sometimes struggle to understand how to motivate and engage educators and students who don't share the same passion [S8].

Moreover, this additional side of research creates more responsibilities for the scientist. Being the lead scientist on a CSG becomes itself a full-time job, inhibiting their personal academic

careers [ELS15]. This includes additional responsibilities for marketing and advertising: normally, scientists don't often need to worry about broadly marketing their research, but for a CSG engaging a wide public audience is essential for adequate quantities of unbiased data [ES1, S2].

Challenges also arise from the discrepancy between the current rationale/logic of academic science and CSGs as game platforms and community outreach projects. Today, the success and careers of professional scientists heavily depend on scientific publications. However, as one scientist explains, this is not always possible with CSGs which mostly have to be designed as long-term projects to build a user base:

> "[I]f we can't get any research results or publications out of it, then we can't put any work into it. So that's typically how it is, projects in computer science are always some kind of research prototypes that are implemented to generate some kind of data or to test the validity [...] And then either the PhD ends or funding ends or something like that and then it's either discontinued or it's somehow taken over as a product [...] by some department that takes care of it. So that's the theory. And the second part never really happens" [DS21].s

The described discrepancy is also experienced in the application for funding. Interview partners expressed the problem that mostly only new, innovative research projects would be funded. However, the realization of a CSG—which could then support innovative research—would require funding to implement the platform, building on existing and well-established solutions. We discuss financial modes of operation more in Sections 4.2 and 4.4.1.

Another major challenge is the different understandings and goals between professional scientists and players. In some of the present case studies, the participants' and researchers' aims do not always align because of the game characteristics which would sometimes conflict with the research goals. For example, the "game" would afford [68, 69] and encourage participants to focus on earning points even when more points would not translate into more accurate or interesting scientific results [ELS15]. This tension has also been observed by Ponti and Stankovic [70] for the case of *Foldit* where certain player behaviors (scripting) produce high-scoring solutions which are not necessarily scientifically valuable [70].

Although developers often emphasized the discrepancies between the scientists' goals and players' goals, it is worth noting here that scientists and developers working on a CSG sometimes also strive for different goals. While scientific accuracy and confidence is most important for scientists, game developers aim to "make [the CSG] bug free" [D22].

Lastly, the issues described so far are primarily concerns for the lead scientist, but what about scientific researchers who are not part of the core development team? There is a need for CSG teams to collaborate with other labs and researchers, yet there exist barriers to those collaborations, as detailed by [ES1]:

> "Whereas right now, in order to do it, you have to know somebody. You have to know somebody who's involved in [game]. Or be willing to send an email cold to someone who's involved in [game], have a conversation about what it is that [...] players can do and how they can help, you know, go through kind of a vetting process, probably go to two or three meetings. And then and only then will your science be ready to submit to [game] players through this rather laborious and time consuming and potentially daunting process. Whereas there is no particular reason why we can't make this information more readily available to the science community so that they can actually do more of it on their own. [...] If [game] is going to be a big resource for the research community, it has to have a wider base of players as well as a wider base of researchers." [ES1]

In summary, being a CSG scientist can be a difficult and full-time job. They are challenged with public-facing communications and education, marketing and recruitment, the academic demand for publications, difficulties funding long-term CSGs, discrepancies between player motivations and scientific goals, and barriers to collaboration with third-party researchers.

## 3.4 Other team members

Within a CSG development team, there are generally three subroles. Software developers create the front-end and back-end technologies for the application and website. Game designers design and implement gamification elements. And community managers form the bridge between developers and participants.

**3.4.1 (Software) developer needs and challenges.** Despite being important workhorses of the team, the developer position is rarely a full-time one—most developers have job requirements besides working on CSGs. In fact, the bulk of software development is often done by students with a professor acting as the project lead. Given the lack of team members described above, there are not enough developer positions to encourage specialization, so most team members work across the full stack of software development.

The first issue developers encounter is the onboarding process. When asked what the onboarding was like, one developer said "None. And terrible" [DS5]. There is a need for documenting development protocols to counteract the steep learning curve for developers, yet developers acknowledged their documentation and documentation process was weak or fledgling.

Because developers are often students or part-time, there is also a high churn rate for CSG developers. This means that there's no guarantee a developer will be able to make productive contributions to the project during their time on the team. In fact, new developers can often weigh down the project by requiring a lengthy onboarding (to what can be a very large codebase and assemblage of operations) and leaving shortly after onboarding. Moreover, because of the lack of documentation, when a long-time developer does leave, the knowledge they gained—e.g., about handling specific bugs or codebase quirks—is often taken with them, lost to the rest of the team.

This high churn rate also seeds distrust among the players when a new developer joins the team; the player community questions what their contribution will be:

> *"Graduate students come in. They work on a project for a year or two and then they leave. And so. [. . .] It's really unclear. Like, are you going to be helpful? Do you care about the community at all?" [DS5]*

Being short-staffed, developers also need to learn new skills and take on jobs outside of their traditional roles—or what they're even qualified to do. Examples include developers as game designers, community managers, artists, or marketers—or vice versa, wherein these positions also require coding expertise (e.g., community managers interfacing with a SQL database) to perform their normal jobs. This is especially true as developers who are not trained in community management describe great difficulties in building and maintaining a community, as described in later sections.

Perhaps the biggest challenge, however, is a fundamental lack of resources. As noted earlier, developer time is limited due to funding restrictions. This has several downstream effects, including a backlog of bugs and a long list of "we should really do these things" [DP13]. This results in bugs only being fixed when they pass a threshold of player complaints [C9]. Eventually, this accumulates into "tech debt"—the developers must deal with old, poorly written code, inflexible systems, and "patchwork upon patchwork upon patchwork upon patchwork as the project had grown [. . .into a] complete mangled mess" [DS5].

This is further exacerbated by a lack of funding to address tech debt; however, this issue will be expanded on in Section 4.2. We summarize two notable effects of the issue of resources: first, developers must often volunteer their time, since there is no budget to employ them for their work. Second, being grant funded, development happens bottom-up rather than top-down: grants fund particular features or datasets, rather than contributing to the holistic design of the game. This lack of overarching vision leads to disparate development movements happening simultaneously. The downstream effect of this is that developers struggle to come to a consensus on the look and feel of the game's design, UI, and onboarding [S8].

Between the bottom-up development culture, grant-based funding, and developer volunteerism, many developers often end up working on whatever interests them or what features are being funded, rather than doing the work that would be most helpful to the project. This easily slips into scope creep as new features get introduced. Yet, these features can easily get abandoned if the one developer spearheading that feature leaves [ELS15]. And this phenomenon is not limited to user-facing tools—software development workflows such as Jira can also get started and abandoned, creating increased difficulty for new developers trying to understand the project as it's spread out sporadically and inconsistently across multiple tracking softwares [ELS15]. In fact, one developer [DGL4] mentioned preferring unfunded work because of how it allows more control over the scope of the project.

Lastly, though not a direct challenge, it's worth noting that there is currently little overlap between CSGs and the gaming industry. CSG developers aren't engaging with the industry, and likewise the industry doesn't recognize citizen science games. Because of this, CSG developers may not be aware of best practices for the design and development of commercial, mass-market games, and professional developers have little to no interest in supporting CSGs—as contrasted with, for example, "indie" development, which veteran developers happily support *pro bono* [e.g., 71].

**3.4.2 Game designer needs and challenges.** Game designers are worth noting for one particular feature: they are never, in our data, their own role on the core team. Despite the gaming industry seeing game design as a wholly distinct position from game programmer, the two are not made distinct in CSGs. Yet, where "game designer" exists as a concept, they have their own role-specific challenges.

The challenges of the CSG game designer can be summarized as a tangle of tensions between the game and the science of citizen science game. There are three components to CSG development: the science, the software, and the game. Science and software can coordinate because scientific software is a common practice, both are familiar with operating on grant budgets and deploying feature by feature, test by test. Game and software are similarly in agreement, since video games are inherently software and game designers and programmers alike are familiar with the fast pace of iterative design and development. Yet, when game meets science, this is where practices diverge. The tension between science and gaming will be further unpacked in Section 4.4.

**3.4.3 Community managers needs and challenges.** Community liaisons, community managers, and other outreach roles form the link between the participants and the developers and scientists behind a CSG. Usually, they monitor all communication media and platforms available for participants to connect with the team and with each other, such as in-game chats and forums. Besides being responsive to the CSG community and forwarding requests and questions from the participants to other team members, community liaisons also translate the needs of the participants for the team: "[T]rying to connect them in understanding, in having the developers know what [. . .] players are really looking for." (C23). In this way, community liaisons can also be understood as the advocates of the participants in the CSG team. At the

same time, they often also communicate in the other direction by taking the science "behind the project and translating that into human language" (C16).

This can also create issues when discussing the realities of science. For example, [ELS15] describes a time when their game partnered with a pharmaceutical company in order to further fund the game, which created vehement pushback and distrust from players. Because the team didn't properly explain the situation or get participant approval, they reflect, the team lost credibility as a non-profit and unbiased third-party working in the name of science. After that incident, they were hesitant to be open about the team's intentions and logistics. "We walk a very fine line between telling them too much and then not telling them enough [. . .] we tend to do that for the science. 'Oh no, we don't tell them about that. Yet'" [ELS15].

Whether due to pushbacks such as that, or other issues causing lack of trust, community managers are encouraged by the rest of the development team to not be fully transparent about the development process. One community manager [C9] described taking a list of bugs to the developers and being told to say the team was looking into it when it was simply low priority:

> "[W]hy can't you [the developers] take ten, literally ten minutes to look at this thing, for example, or just all kinds of small little things like that. [. . .] [W]e had to just, y'know explain it away and go 'oh the developers are working on it.' And [. . .] they're not." [C9]

On the other side, community managers also have to deal with inappropriate behavior from players. In some cases, this means not putting effort into the scientific task in order to simply play the game [DGL4]. In other cases, players will look up the answers to tasks, making benchmarking performance difficult [ELS15]. Ultimately, CSG teams need to be prepared for players to cheat and exploit the game, because it will happen and it will add toxicity to the community [C9] [cf. 72]. According to [C9] and [DGL4], player behavior is driven by whatever is incentivized most by the game system, whether that aligns with the scientific goal or not; it is the responsibility of the CSG team to expect this exploitation and minimize it up front, rather than waiting to address those loopholes.

> "You also have to bear in mind that once there's compensation involved, if people get invested [. . .] cheating will happen [. . .] even though it is a citizen science game, everybody's here to do science together for a higher good. As soon as it's a game, you will get people who want to break the game." [C9]

To make matters more difficult for community managers and liaisons, there is often no designated community manager role. In most cases, the position is part-time or combined with other responsibilities within the team.

Taking over the communication between the CSG team and participants can sometimes be challenging for community liaisons as they have to be responsive to participants but at the same time they are often dependent on the scientists and developers to answer specific questions. Moreover, what has been described as another challenge by team members in the community liaison role is "juggling all the different roles" [C16] which stems from the fact that it is usually not a standalone position but a role integrated into other tasks.

On top of this, the work done by community managers—public communication, mediation, and emotional labor—isn't valued or acknowledged as a skill set.

> "Our biggest weakness, and this is across the board, everybody involved with [game] at all, was our lack of—this is going to sound stupid—customer service skills. [. . .The principal

*investigators] know how to interact with students, postdocs, [etc.] they got that down. [. . .]*
*The students [. . .] know how to interact with other students in an academic setting. But.*
*When it comes to interacting essentially with the general public, we were garbage." [ELS15]*

One participant describes how community managers are treated negatively for relaying negative feedback from the players [C9]: "Asking for bugs to get fixed was wasting developer time or something, like that was the impression that we sort of got [. . .] We bring up concerns hypothetically, but we know that the answer is always going to be OK but [the developers] need to be working on this thing for [the lab] right now." This phenomenon is confirmed from the other angle by a researcher who acknowledges these flaws: "We're just really bad at addressing anything that isn't like the game is broken and it's the end of the world" [ELS15]. As will be discussed in the later section on Science–Game Tensions, the players' needs for bug fixes and experience improvements are treated as secondary to the needs of scientific development.

Where community management exists, these team members are the only bridge between the team and the players. For one case study, community management (when considered as its own job) has historically been a feminine role, entangling gender privilege and power dynamics into the interplay between supervisors, developers, and community managers [C9].

Most developers have no direct interactions with the players [S2, S8, DP13]. In one project in particular, players take on a more driving role in the experimental design, collaborating with the scientists to iterate on the research questions and critically examine the experimental process [DP13]. However, the player perspective is being understood here only from the scientific angle—rarely do the teams directly correspond with players regarding their game experiences.

To summarize, community management is a critical and undervalued responsibility on the CSG team. The struggles of dedicated community managers emphasize that player experiences are treated as secondary to the scientific research, while part-time community managers recognize their lack of skill in adequately communicating with the player base and general public.

## 3.5 Educators and students

Although citizen science games were originally designed for scientific value, and to engage gamers, they have found use in educational settings as well. Several educators have started making use of citizen science games in their classrooms for their value as an interactive learning experience that illustrates concepts with immediate feedback [ES1, E3, E6, E10]. The immediate game feedback is also useful for automated grading, since games come with built-in scoring systems [ES1].

For some, the game aspect of CSGs was also helpful for motivating students [ES1, E7]—as [E7] describes, "When I say game. . . they perk up"—though other educators didn't focus on its gameful nature. Similarly, some educators leaned into the citizen science aspect of CSGs [ES1, E3, E10], using the game as a springboard toward curiosity—discussing how science happens in real life, engaging with current events, and connecting students to a larger community of people interested in science—while other educators focused just on the software itself. One participant [E10] shared a story of having her students watch a video blog and having a scientific question. She emailed the scientist with their question and heard back immediately, giving the students an interaction with a real scientist. "And I thought that was one of the most successful and fun parts of the project," she said.

**3.5.1 Educator and student needs and challenges.** What are the challenges of CSGs in education? First, educators and students are not understanding the game. Partly, this is because

the tutorials are not helpful for them and the game is not well-explained [E7]. Educators also describe how the controls were unintuitive, and it took them a long time to learn how to play—despite their scientific knowledge on the subject, which didn't help. "I did not get the sense that me thinking through the science was going to help," said [E10]. This participant also described feeling personally responsible for their failures in the game, spending as much as several hours on a single puzzle, sometimes needing to "cheat" by looking up walkthroughs or video guides.

Moreover, some educators are skipping the tutorials and/or not playing the game themselves, despite assigning the game to their students, which furthers the lack of understanding. In part, this disinterest in playing is due to the game's design. [E7] commented that there isn't "a good rewarding system," explaining that other games have achievable goals and rewards for getting to the next level. For some games the reward is getting a paper published, "but how often [is that] gonna happen when *I* play?" [E7]. Other times, educators are trying to understand the game but failing or taking a long time to do so. "I'd spend like an hour or two most days, like futzing with the tutorials," says [E10]. These sentiments combined paint a picture that educators are struggling to understand and get involved with the game, which in turn makes it difficult for them to get students interested or help students understand how to engage.

Turning now to the student experience through the lens of educators, the students' lack of understanding comes in part from a hesitation to experiment and a "finicky" nature to the tutorials [E3]. [ES1] said that students have a "hesitancy to try stuff" and a "fear of breaking things." Additionally, other educators mentioned difficulties navigating the user interface and understanding the unintuitive controls [E10]. [E6] notes that this is partly because students come into the classroom "with varying degrees of comfort with technology and varying levels of willingness to experience frustration while they're doing something," a statement echoed by [E10].

And when they are bold enough to experiment, their experiments fail. "The feedback I got from my students is that what frustrated them was when they couldn't get past a level, even after following the directions that were outlined in the wiki and watching other videos of how other people had done it" says [E3]. For [E10], she describes needing to find the "magical combination" of fiddling with the puzzle that solves it. She further notes, to her dismay, that her scientific knowledge as an educator didn't apply much to solve the puzzles, which may be considered a separate issue in integrating education with citizen science games. Even the beginner puzzles are complicated and require trial-and-error, says [E6], "And some of the students find that tedious or frustrating." To resolve some of their frustrations, educators tend to allow students to work in groups or in class with a partner [E3, E6, E10].

Moreover, when given the game during class activities, students ignore the didactics and just play the game as a game. And in doing so, the relative learning gains are inefficient. [E3] comments, "The biggest criticism was that I think because they didn't get a whole lot out of it, that the amount of time that they spent completing the assignment wasn't proportional to what they were getting out of it".

For children ($7^{th}$–$10^{th}$ grade), one educator describes how the students can focus on a game for "maybe 15 minutes" [EP24]. The characters and competition of a CSG are helpful enough to be a change in the classroom, but not enjoyable enough that students would leave their consoles to play [EP25].

Given these challenges, what are the needs of educators and students using CSGs? First, the educators need more control over the game content, making tutorials either more educational or skippable (or both). If the game teaches only basic concepts, the activity will likely not be valuable enough relative to the time it costs to learn the software [E3].

Second, the educators need better support for tracking student progress. As suggested by [ES1], CSGs can be used for automated grading by their interactive nature, but this requires systems that support tracking student progress relative to the learning material, not just relative to the gameplay. As recommended to educational game designers, games that are used in the classroom should be designed with careful consideration of what data are collected by the game and how those data are presented to educators, e.g., through dashboards of clear data visualizations [73]. Moreover, if educational use is something that the CSG supports, then data collection and visualization should be considered throughout the design process, rather than as an afterthought (ibid.).

Third, to better connect with the citizen science value of the game [ES1, E3, E10], educators would benefit from more detail on what science is happening and how it is integrated into gameplay. Fourth, educators reported that students struggled often with technical issues, including institution-specific problems [E3] and issues downloading and installing the software [E3, E10]. Making CSGs more accessible on the hardware available to students and in the classroom (e.g., in web browsers), would greatly increase potential benefits of connecting citizen science games with formal education.

Finally, the greatest need of these stakeholders—which encompasses all prior needs—is that CSG developers need to be collaborating directly with educators. Currently, CSG development is not seeking to meet the needs of educators, and this results quite expectedly in educators not having their needs met when trying to use citizen science games in the classroom.

## 3.6 Other groups

There are several other stakeholders involved in CSGs. This includes funders who support development—both organizational funders, such as the members of NIH, NSF, and private companies and investors who donate to or invest in scientific ventures, as well private funders such as philanthropists and other donors. Another group of stakeholders includes members of third-party for-profit companies within the scientific domain who collaborate with CSG teams on specific projects. Relatedly, there are other companies in the supply chain and market of the scientific domains. For example, when *EteRNA* synthesizes player-made RNA designs, they must interact with companies who produce the scientific equipment and consumable scientific products used in wet laboratory experiments [52].

For environmental citizen science, residents of the environments in question have inherent interest in project outcomes. Lastly, CSGs are influenced by politicians and other policymakers who set laws and regulations regarding science, software, and game development. As discussed previously, we excluded these groups because they do not interact directly with the production or consumption of the player experience. However, future work should examine how these stakeholders further factor into the greater CSG network.

## 4 Overarching themes

After having analyzed the individual stakeholder perspectives on CSGs with a focus on the challenges they face, we now turn to four overarching themes that span across the individual perspectives. These are: [1] roles are ambiguously allocated; [2] limited resources and funding dependencies; [3] need for a CSG community; and [4] science–game tensions. These themes are summarized in Table 3.

### 4.1 Roles are ambiguously allocated

The analysis of the individual stakeholder perspectives has shown that the individual roles within CSG teams are not always clearly distinguishable or distributed and some team

Table 3. Summary of themes across multiple stakeholder groups.

| Theme | Summary |
|---|---|
| Roles are ambiguously allocated | • Developers and participants have vague, overlapping, and/or multiple roles<br>• Overarching visions for the project are blurred by unclear team structure |
| Limited resources and funding dependencies | • Lack of financial and human resources<br>• Long-term maintenance and basic operations are not supported or funded |
| Need for a CSG community | • No centralized community for CSGs |
| Science–Game Tensions | • Issues synchronizing science and game domains |
| — Work Environment | • CSGs developed as scientific software<br>• Emphasis on minimum viable product, not enjoyable experience<br>• Game design assumed to be doable untrained |
| — Communication | • Communication breakdowns between scientists and game designers due to differing jargon and medium of communication |
| — Inherently Difficult Task | • Difficulty integrating science into gameplay<br>• CSG-specific design goals<br>• Requirement of team members to understand multiple domains |

members take over several roles at the same time. Moreover, the case studies examined had different team structures of varying sizes and role allocations. For example, some teams included dedicated community management roles, others relied on combined professional scientists and project lead positions.

CSG teams are an interesting example of team structures and collaborative work because they can be located at the intersection of interdisciplinary scientific research teams and science platforms or more specific game development and game platform teams.

As interdisciplinary teams they combine researchers of a specific scientific discipline such as biomedicine and chemistry with computer scientists and game designers who in general work together towards the joint goal of developing and maintaining a CSG. However, the analysis of the individual perspectives has revealed that there sometimes exist different goals within the teams: whereas the developers are focused on building and maintaining a smoothly working platform and game, the researchers's focus lies on scientific knowledge production and the advancement of research.

> "[T]he most [important aspect] from a developer's point of view, I think, [is] first, trying to make the game bug free and make it easy to play and try to make it fun. I think that might be a little bit different from a scientist's point of view. For scientists is probably the accuracy probably is more important." [D22]

Moreover, the stakeholder analysis has shown that there is sometimes a lack of clarity within the teams about the team structure and the responsibilities that go with individual roles. On the one hand, it remains partly unclear how much of a voice each person gets in executive decisions despite several teams building on so-called "flat" structures within the team instead of hierarchical structures. On the other hand, some team members mentioned their uncertainty regarding who to report to and that they would receive "very mixed messages from multiple directions about [their] jobs" [C9].

These structural ambiguities can make it difficult for individual team members to define their own role within the project. One interviewee described the challenge of "understanding who is responsible for what? Who is capable of what? Where did everybody's role fit in? And then actually define–figuring out my own little space within that" [S8].

Moreover, overarching visions for the CSG in some of the case studies get blurred because of the unclear team structure. There exist conflicting understandings of the vision and current

needs. This goes hand in hand with different understandings of the priorities and therefore leads to gaps between the science and design of a CSG, tensions between different roles within the team and communication problems with CSG participants, whose requests to fix bugs in the game are not being prioritized by the developers. Instead of working on the most urgent issue, developers work on what they prefer to work on. [C9] recalls a time when their team brought on a graphic designer who created showy, ostentatious designs which distracted the team from more important bugs and features:

*"[The team would say. . .] 'This guy made a cool thing. How can we figure out how to incorporate it onto the site?' And it would be like this is literally priority fifty of a thousand. Right now, I don't think we need to be working on this. But that's what would wind up getting worked on or something. [. . .There would be similar "scope creep"] like, oh, what if we made it do this? What if we made it do that? It's just like literally, please, can we just get this done so that we can go back to other things?" [C9]*

In this way, because development is driven by personal developer interest, "a lot of things that happened with [the game] kind of peter off because the leader of that leaves" [ELS15]. Therefore, not only is the scope of the project increasing on unnecessary avenues, work is left half-finished and abandoned.

Another example described by an interview partner is the different understandings of who to hire: while the developers need immediate assistance, and prefer somebody who is "good enough", the project leads want to hire "the best." Without clear team and project structures, setting the same priorities and pursuing an overarching vision becomes largely difficult and working on the CSG tends to move from issue to issue in an ad hoc way.

The lack of an overarching vision is also reinforced by missing resources (see next section). Since funding is mostly grant-based and therefore driven by projects with a fixed time period, the development of the CSG only moves forward when there is a specific grant to fund a specific sub-project, resulting sometimes in a lack of high-level goals for the project.

## 4.2 Limited resources and funding dependencies

No matter how motivated the stakeholders are and how much work they are willing to put into CSGs, the projects always remain strongly dependent on the availability of financial and human resources.

Although, as one of the interviewees expresses in the following, they would continue to work on the projects even without funding, this cannot be sustained in the long run without resources: "I mean there were periods where, where we didn't have funding but that doesn't really matter, we just keep, keep going" [C16].

Hence, the question of resources affects all stakeholders involved, even if not all of them are involved in the acquisition and distribution of resources in their daily working practices. Complaints and concerns about missing resources came up in almost all conversations with CSG team members and can be clustered into a lack of work-hours (developers and their time) and a lack of financial resources.

Most of the analyzed projects faced a lack of team members which would require existing team members to jump from one task to another and to juggle different roles at the same time. Particularly serious for many projects is the lack of developer resources resulting in a backlog of bugs, which get eventually fixed only when exceeding a certain threshold of player complaints. Here it also becomes clear that the participant's requests are not always on top of the priority list of the CSG team:

*"[W]e don't have the manpower to always make everybody happy right now. I wish we did but—and not just manpower but expertise—we only, you know we only have a few people who really know the codebase [. . .] well enough to really develop these things." [S20]*

Although [S20] explains that they "wished" to make everybody happy, the scarcity of resources leads to placing the interests of the CSG team members in the center and the interests of the participants, who devote their leisure time to the project, in the background.

But, as described before, there also exist different goals and priorities within the team which creates challenges when it comes to hiring new developers. The stakeholders responsible for hiring new developers often extend the process to look for the "best," highly qualified candidate, whereas the development team would need and prefer someone good enough who's available. Because of this, [ELS15] describes hiring a developer as "lightning striking" for how rare it is. In reality for the team, however, this process hurts production. Rather than needing a highly qualified developer, the teams just need someone competent who can join quickly and work reliably for a long period of employment [C9] (cf. issues of developer churn mentioned in Section 3.4.1).

Oftentimes, the lack of time and team members or "manpower" [S20] can be translated into the problem of insecure or missing financial resources. Most, if not all, of the projects are mainly based on funding from state agencies, companies, or non-profit organizations. A lot of time must be devoted to writing proposals to obtain grants. One of the biggest challenges, however, is that grants are usually of limited duration and therefore do not ensure long-term sustainability, as one of the project leads explains who had worked on different CSG projects:

*"[S]o there is really not much of [. . .] a very long-term guarantee. [. . .] [T]here is a grant to fund a project, it goes for a few years and then when the grant runs out there is really nothing to support working on a particular project anymore [. . .] [w]hen not some other grants or some other source of funding comes in." [DL19]*

While it is still comparably feasible to obtain funding for new projects, the problem particularly unfolds for long-term maintenance. One of the project leads describes the difficulty of ensuring funding after the initial development phase:

*"[T]here are infrastructure costs, there is still community costs, [. . .] [a]nd we have to do maintenance on the code and [. . .] you know, it costs money to sustain these projects. And I, and I've seen projects disappear like Mark2Cure just because they ran out of funding. So, everybody wants to fund them at first 'cause they are new and innovative but I think the folks who are doing the funding. . .they just don't think about the fact that, [. . .] once the project comes to fruition and it's doing what it's supposed to do, it's not like just because it's successful and effective that it makes money just materialize out of the air. You know, it has to be somehow resourced." [L26]*

Most of the CSGs only develop their full potential in the long term by generating research data. This, however, conflicts with the rationale of funding. One of our interviewees has described this problem, which does not only affect CSGs, as "the sustainability problem in citizen science" (ibid.).

At the same time, funding is often tied to specific scientific projects and established scientific practices such as experiments. As [ES1] states, "developmental resources are placed where people who have money placed them [. . .] money drives a lot of the developments." Most often, this does not include funding dedicated to the implementation of CSGs and the

improvements of the games: "Saying, 'hey, we want to just improve our platform in general, not tied to any specific scientific outcome.' That's hard to get money for" [DP13]. [DP13] explains how the "project lead needs to stretch the scope of a project [in proposing and interpreting grant funding] to cover other necessary features" such as code maintenance. There is often no budget specifically for basic operations, such as code refactoring, bug fixing, porting to other platforms, developing tutorials, or community building. "Like doing just code maintenance is not something you can easily get grant money for" (ibid.). Even something as simple as playtesting for quality assurance is limited on CSG budgets [DGL4, DGL11]. Because of these dependencies, the development of CSGs "is moving at the pace of science funding" [C9].

Due to these difficulties in obtaining scientific grants, some CSG teams additionally turn to other sources of funding as, for example, unrestricted funds of the primary lab or applying for grants that can sneak in maintenance into scientific outcomes. In these practices, development and game maintenance are "wrapped into other things" [DP13]. Another source of funding can be donations, although some of the studied CSG refuse to take donations from the participants themselves: "it's like asking somebody who, like, donates blood, hey could you give us five bucks too?" [ELS15].

The lack of resources also includes a lack of public goods [74] available for CSG development. Namely, there are very few existing code libraries publicly available to assist with the common protocols of citizen science gaming. Because of this, developers have to come up with their own solutions and create code from scratch or build on existing CSGs.

*"You don't really have a lot of people that have worked on similar type of projects so there are not a lot [of] open source tools that you can use [. . .]. So, most of the stuff I had to code like from scratch. You know like the scoring, the [. . .] transitions between images, the [. . .] way the users are giving answers and getting feedback, so all of this needed to be built kind-of from scratch" [D27]*

Moreover, developers are often only part-time or inconsistently employed for the project, so these development requirements are done on a volunteer basis—even players will volunteer their time to assist with bug fixes and needed code changes. This volunteerism-by-necessity leads to developer burnout.

*"I stopped partially because—and this is a phenomenon that comes up, I think, frequently in these projects is that—I just got kind of burned out working on it and I was spending a lot of free time on it without. . ., y'know I was involved in a bunch of different stuff [. . .] And, spending a lot of time on it and trying to help with [tasks] that I didn't really have any experience doing." [DS5]*

In summary, many of the fundamental aspects of game development—fixing bugs, creating tutorials, playtesting, and building a player community—are, funding-wise, afterthoughts partially solved by developer volunteerism, consequently creating burnout.

### 4.3 Need for a CSG community

Several participants discussed the lack of—and need for—a centralized community specific to citizen science gaming. "At the citizen science conferences," says [DS5], "you don't really feel like you belong." He goes on to describe his challenges trying to explain to others what his CSG is doing and why it's intriguing, since few other citizen science projects share similarities with CSGs. And on the other side, CSGs aren't well-established in the gaming industry either [C9]. They are the hybrid of two worlds and supported by neither.

Yet, there are enough people working in CSGs to form a community. The issue is that the field is fragmented and there is little cross-talk between teams. Participants described that, where collaboration or communication across teams existed, only the project lead was involved and acted as a liaison for the group [ES1, S8]. "I've been part of [project] for six years now," says [ES1], "and in that time I've never been to a conference on citizen science or anything even remotely like it. I don't even know what conferences would be worthwhile going to in this field still." Similarly, [S5], [S8], and [C9] acknowledge that they don't have CSG connections outside of the team. It's "the sort of stuff that the group leader engages in," says [S8].

"I think that the community being fragmented hurts all of us." [ES1]

Participants speculated on what value a CSG community would provide. These values included simply having someone else thinking about these issues (such as design issues) to talk to [DS5], discussing pedagogical methods and tutorial design [S8], as well as sharing information, supporting each other, and talking about bigger problems [DP13]. For example, [DP13] suggests it is important to have ethical discussions as a community about the roles of the CSG player, drawing from [75]. He goes on to consider the idea of stakeholder meetings for the community as a whole (i.e., stakeholders of the field of CSGs).

## 4.4 Science–game tensions

There are three broad disciplines which work together to create CSGs: science, software engineering, and game design. Software engineering naturally pairs with both science and game design, by the fact that both science and game design fields involve building software. However, science and game design do not mesh as easily, and tensions exist when these two fields "interfere" [76].

This is not the first article to notice this phenomenon, but it may be the first to state it as a team-based dilemma. Ponti et al. [46] identified tensions between the values of open science and the gameplay of secretive competition, a symptom of this broader dilemma. This tension was confirmed by Miller and Cooper [60], attributing this issue to the lack of people who specialize in both the scientific topic and game design, as suggested by Prestopnik and Crowston [77].

But is the problem truly that CSGs need developers specializing in science and gaming? What makes science and game design seemingly incompatible? What is it that creates, as [C9] puts it, a "fundamental push-pull between [. . .] the science and the game part in 'citizen science game'"? We further unpack this tension—as a team-based tension—by highlighting three aspects of this dilemma: the work environment, communication, and the inherently difficult task of CSGs.

**4.4.1 Work environment.** First, the development of CSGs is more aligned with the work-flow of scientific research than game development—within development teams, CSGs are framed and developed as scientific software instead of as games. What do we mean by this framing? Game studios operate on tight iterations and frequent releases, while scientific teams are funded by much more long-term grants for very targeted applications.

*"[Scientists are familiar with having] milestones and objectives and stuff like that on paper, but having no practical, real world experience from a program development perspective for, like, how to do this in time for a consumer audience to be happy, because it's not something that's familiar in the sciences, because you're not answerable usually to random humans. You're answerable to these six months, 12 months, five year plan [. . .] 'oh, you know, we'll just plug along at this thing and, you know, we do it when we have the money and we don't do it when we don't have the money [. . .] like, OK. But actual game studios* cannot *survive that*

*way. . . The fact that where I work is essentially a game studio that hasn't released a new title since [approximately ten years ago], and we still exist—in game studio terms, that would be absurd. [. . .] There's that disconnect between knowing what it takes [. . .] to sell a game to a user base. And how that doesn't jive very well with laboratory workflows." [C9]*

In essence, CSGs rely on scientific funding mechanisms rather than game-based funding, and this has downstream effects for how development happens and what gets developed (refer back to Section 4.2).

The framing of CSGs as scientific software is further confirmed by the project lead's positioning. [DS5] describes that it's not unusual for the project lead to be a scientist and thinking about the project scientifically. [DP12] also notes that their project lead didn't understand the software language they were using for visual display and animation, or that they even understood games. Because all of our case studies start from scientific problems, it makes sense that the project lead would be a professional scientist. Yet, this positioning does not set the project up well to handle the other aspects of CSGs like game design, game development, or public communications.

Finally, the framing of CSGs as scientific software has the downstream effect which we term *polish versus possible*, described at length by [DS5].

*"There's a difference between creating the software to make it* possible *[their emphasis] to do something and creating the software to make it easy to do something. They [the development team] were good at the first thing, creating it possible, right. But not good at [making it easy]." [DS5]*

The scientists on [DS5]'s project were always on the cutting edge. This is the work mindset they understand: solve new and interesting problems, that's what the funding is for anyway.

*"They're trying to figure out: how can we make it possible for people to solve [the latest research question]. That's an important question. But then* all *of the other stuff they've already done, that's in the past. That's not the cutting edge of research anymore." [DS5]*

Every feature they develop, once made possible, is never polished to be user-friendly (cf. [S20]'s statement that although they can't "make everybody happy," they focus on the happiness of the scientists). For example, players have complained that new features are being added while major bugs are still present.

*"I'm trying to find a polite word to say, but it's just [. . .] detritus. You know, like it's just kind of, we've already done that. And that's just out there. And if people want to work through it, they can work through it. But right now, we're focused on [. . .] the next research angle." [DS5]*

While CSG players don't directly feel the effects of funding sources or project management, they do feel the effects of polish, or lack thereof [60]. CSGs as scientific software are grating, frustrating experiences when players attempt to use them as playthings rather than serious tools, because they were developed with the mindset of creating serious tools, not games. Though we leave this as future work, it is worth exploring in what specific ways the development processes of CSGs differ from projects that are purely scientific or purely games.

The last issue regarding work environments affecting science–game tensions is that CSG teams largely underestimate development of the game aspects. [S8] recounts her experiences with developing for CSGs:

*"What I personally have learned is kind of the complexity of what goes into a good game and the importance of, y'know, I didn't understand the importance of writing good quality code until I started writing games, actually, and I've written a lot of code in my time [. . .] What I really learned was kind of the value of understanding the story that you're trying to tell and taking into account people's ideas and testing and doing all of that. And I think a lot of this is very obvious to a seasoned game designer. [. . .] Sitting there going 'Ah, game design, how hard can it be?' And then you sit there and you're drowning in code and you have a game that isn't fun." [S8]*

Even for more experienced game developers, the game aspects of CSGs are still underestimated. [DGL11], for example, emphasized the challenge of designing levels and their pacing. He describes excessive production on his team due to underestimating how much balancing and art would be required for the number of levels they wanted to put into their game. Similarly, [DGL4], in reflecting on his past work, speculates that many aspects of his game could have been better if the team knew more about game design.

Perhaps it is a property of game design that this type of expertise appears easy to replicate. No one would expect to be able to have an intuitive knack for molecular biochemistry or be able to simply "whip together" something in quantum physics. Yet, developers consistently make this assumption of game design—and, to a lesser extent, instructional design (with respect to developing tutorials and educational resources).

Icons from The Noun Project licensed under CC BY 3.0: science by Saideep Karipalli; code by Evon; controller by Abdul Karim; marketing by Tri Sudarti; community by Alzam; translator by Lutfi Gani al Achmad.

**4.4.2 Communication.**   The second aspect of science–game tensions is the communication breakdowns that happen within the team. Refer to Fig 1 for a diagram of the typical channels of information dissemination to the CSG team and its participants.

According to [S2], "[t]he biggest wall is between the communication of the scientists and the game designers." [ELS15] similarly mentions that communications failed most often with design; for example, a scientist would describe a feature to the developer and their meeting would end with a goal in mind, then a few days later the developer would show off their work and the scientist would say "oh yeah, no, that's not what I meant".

Similarly, developers struggle to communicate software capabilities and design intentions to the scientists—meaning communication is failing in both directions [S2]. In this way, there is a need for a "middle person" to translate ideas between the scientists and the developers and designers (ibid.). [ELS15] similarly describes becoming the "intermediary" with other collaborators, extending the "middle person" role to one which interfaces even outside the team.

Another aspect of communication is whether discussion is online or in-person. [ELS15] believes that having in-person meetings overcomes many differences in jargon: "if you don't get something or if you misinterpret something, you know, in the next minute, we're going to figure that out." Emails, on the other hand, can be misread or misinterpreted, and Zoom meetings are easy to "tune out" (ibid.).

In short, much of the communication within CSG teams is about coming to a shared understanding for its design. This task is made difficult both by the medium—when communication happens online—and by differing sets of jargon, e.g., between scientific terms and game design terms, or even between two scientific backgrounds with differing epistemologies. [S2] gives the example of "model" referring to a cognitive model or a mathematical model depending on one's background, which can lead to different interpretations of what's being discussed. Having someone who understands multiple backgrounds and can relate ideas is helpful for mediating discussion in these strongly interdisciplinary teams.

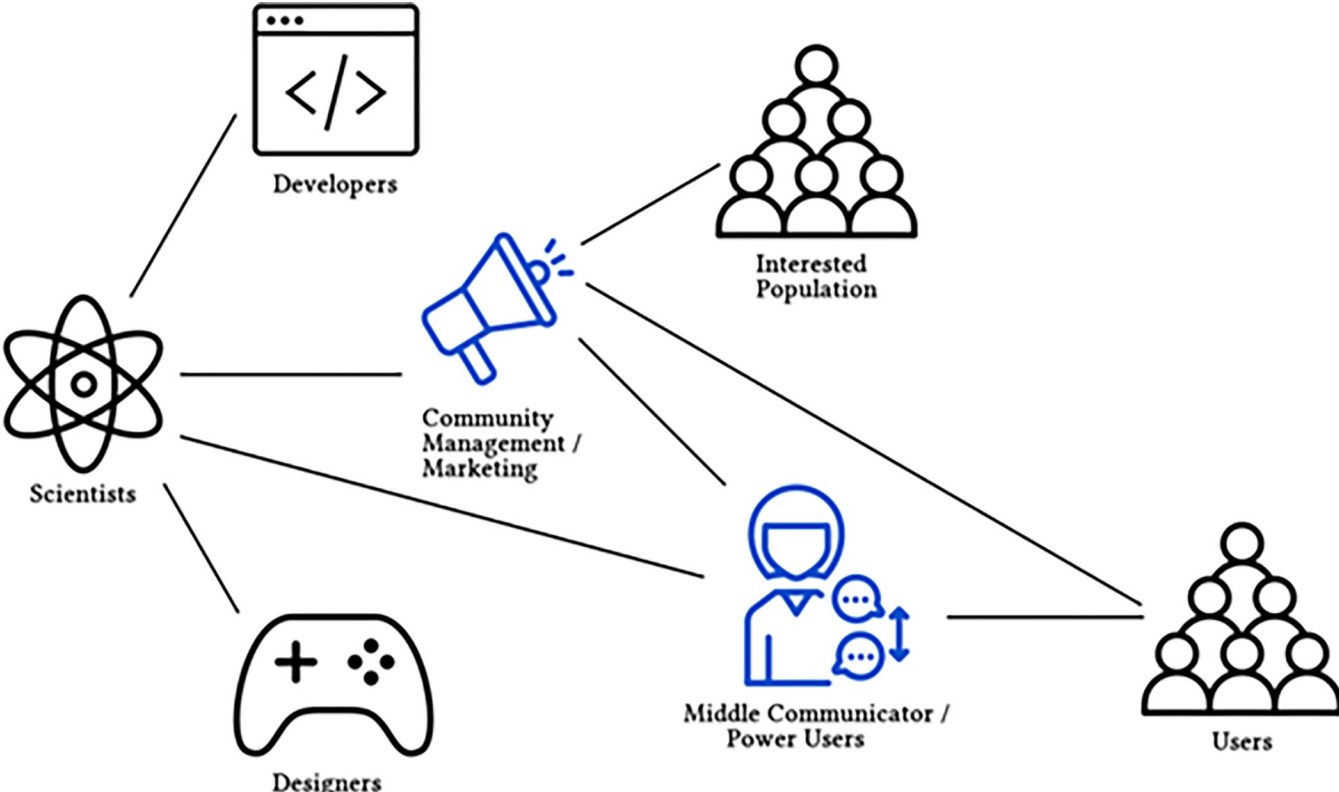

**Fig 1. A diagram of typical channels of communication and information dissemination.** Roles in blue are sometimes absent, in which case limited communication may pass through directly (e.g., from Scientists to Users) or be absent.

**4.4.3 The inherently difficult task.** Finally, making a CSG is inherently a difficult task because it requires integrating the strongly disparate fields of science and game design. [DGL4] notes that the choice of both the scientific domain and the task are critical for designing a good CSG—some domains and tasks lend themselves well to a game approach, while others are much harder to make into a game. The more specific one's task requirements are, the harder it is to integrate into gameplay because it becomes a rigid requirement, "You're basically being handed a [game] mechanic" (ibid.).

Additionally, CSGs come with more design goals than commercial games. CSGs need to design for data collection, data logging, and making clear to players what gameplay counts as scientific contributions. When making the tutorial, developers struggle to teach a domain that's still being developed—it can be difficult to know what aspects of the game will be important after several years, meaning that the "curriculum" of the tutorial might change [DS5]. Notably, some data-centric [78] citizen science tasks have seen success in being a mini-game within a larger, successful commercial game [19, 79, 80].

The problem of citizen science requiring interdisciplinary expertise has been known for several years. Miller-Rushing describes how citizen science projects require "expertise in science, working with volunteers, education, technology, translation of science to policy—that can be tough to bring together" [81]. But for CSGs, they not only require diverse skill sets but an integration of those skill sets. It's not enough for each team member to take on a skill set, they must share an understanding of the other domains.

[DS5] reflects on how important it was for the software developers to understand the game; it was okay for them to not understand the game if they were focused entirely on code optimization or other software-only tasks, but developing for either the science or the game aspects required understanding other domains. This is especially true for the project leads who need to understand how audiences will respond to a product [C9].

For many scientists and developers getting involved in a CSG, this is their first experience with game development, and their mindset is often still focused on the scientific approach [DGL4]. "This is my first real experience doing [UI design]," says [DS5], "And it was like, [people] do not behave at all like you would expect [them] to." Similarly, [S2] recalls working with the scientists to create game design documentation, mentioning how this act annoyed the scientists because it was too much non-science work for them. Moreover, game design is more challenging than teams expect. Even for teams with game design experience, they still underestimate the amount of effort needed to produce a good game experience [DGL11].

What happens when you have a design task that requires both game design and scientific expertise but don't have experts in both domains collaborating? In short, the result is not fun. "I'm afraid that people are actually starting to relate citizen science games with games with no fun. That is what they already do with educational games," says [S2]. When asked how to fix this problem, they suggested: "A scientist should know that they know about science, but not about design or about development. So getting associated with a group of designers and developers is really important".

Despite science and game design being often in opposition, a good CSG requires them both —combining both science and gameplay "in a way that is more natural and every part of the game feels compelling" [DGL4]. If CSGs are to become better in the future, this will mean changing the work environment and the communication patterns to better support the inherently difficult and inherently integrated design task.

## 5 Takeaway recommendations

In this section, we review 17 Takeaway recommendations for CSG developers. These takeaways are given to us by our participants as well as our own recommendations which come from a synthesis of our findings. For other recommendations on CSG development, we refer to previous literature [60, 82, 83].

First and foremost, we and our participants strongly recommend **(T1) learning from what has been tried in CSGs—and the lessons learned—*before* creating a new CSG**. [DP12] says, "There are so many other projects who have actually a lot of experience about gamifying a scientific subject. I would tell them [a new team / new CSG project], go interview those guys, learn from them before you start up. Because many people have been through a number of those things." Similarly, [S8] says, "Understand the literature better [. . .] not even with regards to reading the papers [. . .] but understand what different projects have done. Understand sort of where your work fits in with that. . . There are a bunch of different kinds of citizen science games".

To this point, it is critical to **(T2) reflect on whether building a CSG is the right approach for your project**. CSGs are expensive and time-consuming. Although they can be powerful, they solve a very specific kind of problem, one which is not suited for every project.

*"If you're thinking of doing a citizen science game, you should understand that and then decide if you have the wherewithal to do that. Because there are a lot of ways to motivate engagement in citizen science and games are just one of them. And a lot of the times the citizen science project can be quite successful even if it's not a game project. [. . .] A game could be a*

*good choice, but it's more work than you might realize. And particularly to do it well, and particularly if you want the game to attract people you wouldn't otherwise attract another way."*
*[DGL4]*

CSGs require a lot of time, money, and expertise. If you are planning on making a CSG on a low budget without the dedicated expertise in science, software, game design, and community management, we recommend you consider other formats for your project. If you do have the resources to make a CSG, first prototype the concept and build on what other CSGs have done before committing to the project in full. **(T3) Approach CSG development with resources and a plan**. See [60] for further discussion of funding opportunities for CSGs.

Keep in mind that **(T4) some domains are more easily gamified than others**:

*"If I had a choice about context, I would look really closely at contexts that I thought had more gameplay opportunit[ies] in them. So some of those contexts, I think, like astronomy, I think lends itself well to games. There's a lot of science fiction games that you can do interesting things with astronomy. There is, I think, some kinds of scientific work that lend themselves well to games and other kinds that might be a little more challenging. And I might try to set myself up for success by focusing on a context that seemed like it would work well with games." [DGL4]*

Throughout the development process, there are several aspects to focus on. **(T5) Outreach, such as explicit advertising, is critical for building a player community**. In addition to social media, stakeholders described promotion from influencers (especially non-gaming influencers) and getting on a morning television show as very successful outreach initiatives for their project.

**(T6) Focusing on retention** and the **(T7) player community** is also critical for long-term engagement and, ultimately, the health of the game. "Player retention is paramount" says [DP12]. Communicating clearly, regularly, and transparently with your users is key for building community. Being transparent with the community—about decisions made behind the scenes and how the science is done—is critical in order to avoid "sound[ing] too promising" [S28] and ensure that participants support, or at least understand, developments of their CSG.

*"Focus on the community. That's what will make [or] break your project. Give them everything you can think of to help them be successful. Listen to their feedback on what they want. Let them surprise you with their out of the box thinking and nuanced realizations and considerations. Communicate often. Make sure that they're able to communicate and collaborate among themselves. Keep a close eye on the pulse and health of the community." [DP13; post-interview comment]*

With respect to the development team, **(T8) ensure that everyone understands the project** and **(T9) their role within it**. Most team members will need a basic understanding of the science, technology, and game mechanics in order to collaborate effectively. [DGL11] emphasizes, "Really understand the science. Really understand the technology and mechanics."

**(T10) Use each team member's expertise to your advantage**. [C9] says, "Be willing to have the scientists in your projects defer to the game designers about matters of game stuff. And make sure to have the game designers be totally clear on the scientific priorities and what has to actually come before game design."

For development itself, [DP13] says, "Don't let tech debt accumulate." CSGs, which can be decade-long projects with over fifty developers working on them throughout the life of the

project, are susceptible to becoming overly complex or hacked-together unless conscious work is put into keeping the code and workflow clean and documented. Additionally, [DS5] adds that "Having players involved in development is really, really good [. . .] immensely valuable for the research." Player input during design and development can not only help identify user experience problems that the team may have missed, but also provide insights into how players interpret mechanics, mentally model their understanding, and engage with the CSG. These insights are invaluable for bridging the gulfs of execution and evaluation between the player and the game [84].

Ultimately, if you are working on a CSG that you expect to last for years, we recommend **(T11) taking the time and budget now to improve your technology, workflow, developer onboarding, community management, player interactions, and so forth**. This polish will improve your scientific output in the long-term [60].

Ongoing maintenance is easier if you start with a strong concept. To this, [DP12] and [DP13] suggest **(T12) putting first and foremost the gamification** (how you gamify the task) **and the abstraction** (how you represent the task to an unfamiliar audience). "Players should be able to come in without any knowledge" [DP13].

Supplement the abstraction with features that let the community teach each other. Since CSGs are, typically, social projects by nature, **(T13)** "**design ways that your community can interact**" and "**facilitate knowledge transfer**" [DP13]. This will improve the flow of knowledge from expert players to novices, but also from players to developers. As [ELS15] says, "It's all about communication."

What you're trying to do, according to [DS5], is create a scientific community—not teach students in a classroom. This difference affects how you approach the curriculum design, attempting to **(T14) foster open scientific engagement rather than overwhelm them with technical knowledge**.

In this way, **(T15) the polished experience matters**—"entertainment really matters" [DGL4]. This was also found by Miller and Cooper [60], who identified game polish as a significant factor for CSGs. Yet, in contrast, we found this perspective highlighted only by developers whose role included game design; other team members were more focused on the scientific goals of the project and less concerned with the player experience. This recalls the science–game tension and highlights the unclear vision of CSGs. For CSGs to be widely accepted as enjoyable experiences, though, we agree with [DGL4]: the player experience must be enjoyable for the game to serve its purpose as a leisure activity that engages a wide audience.

Lastly, if you do make a successful game, [DGL4] warns against a common pitfall when publishing about it: don't let your research be simply "marketing talk" about how great your game is. Instead, we recommend to **(T16) focus publications on lessons learned and general takeaways that other projects can benefit from**. Contribute to the greater CSG community by open-sourcing your work, creating a platform for other CSGs, or providing other resources for getting started.

Finally, **(T17) help develop the CSG community** by collaborating with other CSG developers, such as through the CSG developer Google group: https://groups.google.com/g/csg-developers.

## 6 Discussion

In this article, we conducted a joint qualitative analysis of 57 interviews with stakeholder groups from 10 different CSGs to understand their differing individual perspectives and needs (Table 2) as well as shared and/or cross-cutting challenges, namely the ambiguous allocation of roles, limited resources and funding dependencies, the need for a citizen science game

**Table 4. List of takeaway recommendations for CSG stakeholders for CSG project success.**

| Takeaway Recommendations for CSG Development Teams |
| --- |
| **(T1) Learn from previous CSGs**—understand the successes and failures before creating a new CSG. |
| **(T2) Consider whether a CSG is right for your project**—CSGs are expensive and time-consuming and solve a specific kind of problem; they may not suit your project. |
| **(T3) Come in with resources and a plan**—A CSG without secured funding and experts will likely result in failure. Once you have your resources, prototype your technology and build on what other CSGs have done so you don't have to reinvent everything yourself. |
| **(T4) Focus on domains with gameplay opportunities**—Some domains are more easily gamified than others. |
| **(T5) Focus on advertising**—outreach is critical for building a player community. |
| **(T6) Focus on retention**—keep players engaged long-term. |
| **(T7) Focus on community**—communicate often, thoroughly, and transparently with users; create a community. |
| **(T8) Have a shared understanding**—most team members will need a basic understanding of the science, technology, and game mechanics to collaborate within the team. |
| **(T9) Give clear roles**—make sure everyone on the CSG team understands the project and their role within it. |
| **(T10) Defer to the experts**—Make sure game designers are clear on the scientific priorities (what has to come before the game design) and have scientists defer to game designers on matters of game design. |
| **(T11) Control various forms of debt**—Don't let technical debt accumulate, continuously clean the codebase; continuously improve your developer onboarding, workflows, community management, player interactions, tutorials, etc. These forms of general maintenance will have long-term benefits for scientific research. |
| **(T12) Consider abstraction and gamification first**—Take time at the beginning of a project to develop a strong idea of how you will gamify the task and represent it to an unfamiliar audience. |
| **(T13) Design for socialization and learning**—Build features to allow the community to interact and facilitate knowledge transfer. |
| **(T14) Develop curiosity over education**—Aim to build a scientific community rather than a classroom; it is better to have someone engaged and interested in learning more than to dump knowledge on them from the start. |
| **(T15) Polish really matters**—Game polish is critical for engaging players. |
| **(T16) Publish general takeaways**—If you do make a successful CSG, share lessons learned and general takeaways that other projects can benefit from; consider publishing code libraries or open-source resources for other projects to use. |
| **(T17) Help create the CSG community**—Collaborate with other CSG developers; consider joining the CSG developer Google group: https://groups.google.com/g/csg-developers |

community, and science–game tensions (Table 3). Here, we connect these findings to prior work and derive recommendations—both from our research and directly from our participants—on how identified issues can be addressed (Table 4).

The issue of ambiguously allocated roles mirrors findings in Science and Technology Studies [e.g., 85] and by Wudarczyk et al. [86] that interdisciplinary teams need alignment on project expectations, a common goal, an understanding of different practices, agreement on terminology, establishment of shared knowledge, transfer of essential technical knowledge, and embracing diversity as an asset.

Counter to Golumbic et al., who found that scientists enter citizen science for funding reasons [39], we found that there is very little funding available—relative to their needs—for CSGs. This may be partially due to a 'cooling' of funder enthusiasm in the past 5+ years, as CSGs have become less novel. The 'lumpy' grant-based model of CSG development creates challenges for ongoing basic operations of game development, such as bug fixes and code refactoring, tutorial development, playtesting, and community development. Consequently, developers volunteer their time to these basic operations, leading to burnout and churn, which in turn exacerbates the problem, for example, as many developers enter the project briefly then leave, creating a messier codebase.

A third theme was a need for a CSG community. Several problems with CSG development currently could be mitigated with more shared knowledge, code, and other reusable resources.

Despite the existence of communities for citizen science and for games, there is no community for citizen science games. This fragmentation means that most CSGs are starting from nothing every time. There is still no general solution for developing CSGs—most projects develop and grow dynamically, despite platforms such as SciStarter (https://scistarter.org/) to promote and support projects. As one possible first step toward a global CSG community, the authors have put together a Google group for CSG developers (https://groups.google.com/g/csg-developers), with 55 members at the time of writing. We encourage anyone interested in CSG development to join this group and collaborate toward other ways of centralizing the community.

Fourth, we expand on prior observations of the science–game tension in CSG [46, 60, 70]. We break this tension down into several aspects: CSGs are developed as scientific software, development focuses on the possible not the polish, there are communication breakdowns between scientists and game designers, and implementing scientific tasks into gameplay is inherently difficult.

Notably, these four challenges interact and compound. Implementing a scientific task into gameplay may be hard, but it's even harder when there is no dedicated game designer on the team and developers are ambiguously in charge of design and development. These ambiguous roles are strained more by not being funded to work on them—with no funding to guide their efforts, they focus on what interests them, resulting in half-finished projects abandoned when they burnout and leave. Their absence impacts the player community, who gains distrust for new developers coming in, questioning whether they will stay and help or create a mess and leave as well.

Communication especially is an intersecting factor across many themes and individual challenges. While we focused on communication only with respect to the science–game tension, in many ways, communication is the core problem of ambiguous roles, building a CSG community, and building trust and competence with the CSG players.

How should these issues be resolved? Is it really true that science and gaming are epistemically opposed? We argue that the issues with CSG development are not unsolvable. Below, we list recommendations gathered from our interviewees and synthesized from our analysis. While these recommendations are not a panacea for the problems in CSGs, we hope that this can be the beginning of a conversation which recognizes these problems and actively works to address them—together, as a global CSG community.

## 6.1 Limitations and future work

As stated previously, although this work set out to explore stakeholder perspectives on CSGs, we did not analyze all potential stakeholder groups, excluding funders, policymakers, and companies in the broader supply chain of CSGs. This was both because these stakeholders do not directly contribute to the production or consumption of the CSG player experience, but also a decision of scope for the purpose of this project. Therefore, future work could examine these perspectives.

Second, while we thematize our participants' descriptions of their experiences to the best that we understand them, it remains possible that we have missed or misinterpreted some aspects of their perspectives and our representations must always remain interpretations. Future research can validate this study through additional analyses, longitudinal studies, or other methodologies, as well as empirically testing the recommendations provided for efficacy.

## 7 Conclusion

CSGs have the potential to improve scientific research and public participation in research. To realize that potential, the diverging needs and tensions between their diverse stakeholders

need to be addressed. In this article, we outlined the needs and challenges of several key stakeholder groups, including the ambiguity of roles, limited resources and funding, and tensions between the fields of game and science. Our hope is that this work begins a conversation on how CSGs can be more than novelties and create lasting, mutually beneficial collaborations between researchers and the public.

## Acknowledgments

The authors would like to thank all of the stakeholders who engaged with this work and with various CSGs. Together we will make great strides in scientific research.

## Author Contributions

**Conceptualization:** Joshua Aaron Miller, Libuše Hannah Vepřek.

**Data curation:** Joshua Aaron Miller, Libuše Hannah Vepřek.

**Formal analysis:** Joshua Aaron Miller, Libuše Hannah Vepřek.

**Investigation:** Joshua Aaron Miller, Libuše Hannah Vepřek.

**Methodology:** Joshua Aaron Miller, Libuše Hannah Vepřek.

**Supervision:** Sebastian Deterding, Seth Cooper.

**Writing – original draft:** Joshua Aaron Miller, Libuše Hannah Vepřek.

**Writing – review & editing:** Joshua Aaron Miller, Libuše Hannah Vepřek, Sebastian Deterding, Seth Cooper.

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
