## [Decision Letter · Decision Letter 0]

13 Jan 2023

PONE-D-22-30825Practical Recommendations from a Multi-Perspective Needs and Challenges Assessment of Citizen Science GamesPLOS ONE

Dear Dr. Miller,

Thank you for submitting your manuscript to PLOS ONE. After careful consideration, we feel that it has merit, however, there are some small improvements that could be made. Therefore, we invite you to submit a revised version of the manuscript that addresses the points raised during the review process.

 Both reviewers were highly complementary of your manuscript, and have provided some suggestions to improve your manuscript. I invite you to carefully consider the reviewer comments below so as to improve your manuscript before acceptance and publication.

We look forward to receiving your revised manuscript.

Kind regards,

Peter Edwards

Academic Editor

PLOS ONE

Journal Requirements:

5. We note that Figure 2 in your submission contain copyrighted images. All PLOS content is published under the Creative Commons Attribution License (CC BY 4.0), which means that the manuscript, images, and Supporting Information files will be freely available online, and any third party is permitted to access, download, copy, distribute, and use these materials in any way, even commercially, with proper attribution. For more information, see our copyright guidelines: http://journals.plos.org/plosone/s/licenses-and-copyright.

Reviewers' comments:

Reviewer's Responses to Questions

**Comments to the Author**

1. Is the manuscript technically sound, and do the data support the conclusions?

Reviewer #1: Yes

Reviewer #2: Yes

2. Has the statistical analysis been performed appropriately and rigorously? 

Reviewer #1: N/A

Reviewer #2: Yes

3. Have the authors made all data underlying the findings in their manuscript fully available?

Reviewer #1: No

Reviewer #2: No

4. Is the manuscript presented in an intelligible fashion and written in standard English?

Reviewer #1: Yes

Reviewer #2: Yes

5. Review Comments to the Author

Reviewer #1: Overall this is a very readable and useful account of citizen science games! I am so excited to see it! Here are some comments:

1. In the introduction, you mention that there are some studies that identify challenges with using CSGs. Another one to consider: They aren't engaging/well-designed sometimes. in comparison to typical popular games, they may not be as fun to play. (Cite Knowledge Games by Karen Schrier?)

2. Ethical considerations of citizen science games were discussed, but this article on that specific topic was not cited: https://journals.suub.uni-bremen.de/index.php/gamevironments/article/view/147/131 (The ethics of citizen science games)

3. Why did you use purposeful sampling? Was this because of the small sample size of CGSs or ones you have access to?

4. Was there just one person conducting the thematic analysis or was this a shared task among the researchers? It's stated but still not clear why the secondary analysis was used. Themes vs. topics -- can you give examples of what you mean by this?

5. co-laborative  collaborative? Or is this a term that is undefined/jargon that should be defined further?

6. It's not clear why the thematic analysis was used in the first place, and it is not clear enough why the second series of methodologies were used.

7. This sentence, and the surrounding sentences, are a bit confusing. "Following an ethnographic and grounded theory approach data collection and analysis did not present independent successive but alternating phases." It's also confusing that you have some games used in the first approach, and then a subset in the second, and why it was a subset. I find the methods descriptions and rationales confusing and glossed over too much.

8. I am glad you explained the games. Should that be moved up earlier in the methods section? Could there be a chart of games and which methods you used with which?

9. The results in general are well-organized. I don't think any of the results are particularly surprising--they all make sense--but it's good to have them be written out. I also agree that people seem to have a hard time realizing the distinction between game designers and developers. This feels like part of the issue/tension that exists with CSGs being outside of the realm of the industry/professionals. A game designer of a CSG needs to not just be a professional designer but really understand CSGs and how to design them--the science, the community, and the stakeholders. This is a difficult and complex skillset to find but it should be found and not given to the game developer (no, please, no) or the scientists/grad students (also, no). Game design is a profession that should be valued (same with community managers, etc). As for educators/students, that part of community building and education needs to be established from the start.

10. The ecosystem among everyone needs to be valued from greater society--I agree!

11. "At the citizen science conferences,” says [DS5], “you don't really feel like you belong.”--This is very important! It's such a hybrid, new, interdisciplinary approach that you feel like you are not anywhere and there is no broader professional community or set of practices that you can access. It's also extremely difficult as a person of color/woman/other marginalized group even further.

12. Recommendations are great to see. I would include some with valuing the professions of game design, development, and community management more readily, with a plan of how to fund these as a professional, rather than through student work or interns. Can they get funding for a vendor? Also, I know this is an area of expertise for some of you, but it is problematic when it says "gamified" throughout the paper-- these are games (CSGs) not gamified content. To me, they should be considered full-fledged games from the outset.

13. I like the table for recommendations. Could you separate it out as recommendations for development team, scientific community, educational community, etc?

Thanks again for writing this article!

Reviewer #2: This is a thorough, and very well presented longitudinal study of Citizen Science projects employing gamification techniques. I thank the author for a very well written paper, that will contribute a lot to the understand of stakeholder views, challenges and issues long into the future. I recommend that this work is accepted.

My only comments would be:

1. A more thorough description of the projects, with an explanation of the type of gaming employed (leader boards, collaborative, first-person, exploration etc.), and perhaps an investigation on how these types correlate to the opinions of stakeholders - this could be a whole follow-up paper however!

2. Perhaps a section on the disciplines involved in each project, how they relate to the school subjects, and if that correlates with any of the opinions of the students and/or teachers involved.

3. There are now some very strong examples of citizen science being incorporated into commercial gaming, instead of the other way around (see MMOS, Project Discover on EVE online https://www.eveonline.com/discovery, and Borderlands 3 https://www.pcgamer.com/how-eve-online-and-borderlands-3-merge-citizen-science-and-minigames/) - where the game design and creation very much takes centre stage. This probably should at least be discussed at some point.

6. PLOS authors have the option to publish the peer review history of their article (what does this mean?). If published, this will include your full peer review and any attached files.

Reviewer #1: No

Reviewer #2: No

---

## [Author Response · Author response to Decision Letter 0]

26 Jan 2023

Thank you very much for your thorough reviews. We hope we have addressed all of your comments and list our changes here:

R1: In the introduction, you mention that there are some studies that identify challenges with using CSGs. Another one to consider: They aren't engaging/well-designed sometimes. in comparison to typical popular games, they may not be as fun to play. (Cite Knowledge Games by Karen Schrier?)

We added to the introduction that CSGs struggle with fun, engaging play, citing Schrier as suggested.

R1: Ethical considerations of citizen science games were discussed, but this article on that specific topic was not cited: https://journals.suub.uni-bremen.de/index.php/gamevironments/article/view/147/131 (The ethics of citizen science games)

We added citations to Schrier on the ethics of CSGs as suggested.

R1: Why did you use purposeful sampling? Was this because of the small sample size of CGSs or ones you have access to?

Our aim was twofold. For the HCI study, we aimed to gather a representative sample from among many of the most popular and well-known CSGs. This was, as you say, limited by who we had access to speak to, but we sought to sample from multiple games without sampling too heavily from any specific one. For the ethnographic study, we took a deeper approach to ensure that our findings are grounded in lived truths. However, no change was made to the text because we believe this is adequately captured in the nature of purposeful sampling.

R1: Was there just one person conducting the thematic analysis or was this a shared task among the researchers? It's stated but still not clear why the secondary analysis was used. Themes vs. topics -- can you give examples of what you mean by this?

We added a clarification that the initial thematic analysis was performed only by the first author.

As described in the paper: the initial thematic analysis of the HCI study was leading to topics rather than themes, which is why the joint analysis was performed (to produce themes). And, as stated in the first line of Section 2.3, the joint analysis was performed by the first two authors. Regarding topics vs. themes, we added a note to refer to Braun and Clarke 2019 which helps disambiguate this: topics are data domains (e.g., “Scientists’ Perspectives”) while themes are “patterns of shared meaning underpinned by a central organising concept” (Braun and Clarke 2019, p. 1). Our four themes in Section 4 are examples of this.

We added a clarification on the discussion of topics vs. themes to refer to the works of Braun and Clarke.

R1: co-laborative  collaborative? Or is this a term that is undefined/jargon that should be defined further?

We added quotes around “co-laborative” with a brief definition to accompany the cited reference.

R1: It's not clear why the thematic analysis was used in the first place, and it is not clear enough why the second series of methodologies were used.

We added further explanation to the purpose of the first thematic analysis and to the purpose of the second qualitative analysis.

R1: This sentence, and the surrounding sentences, are a bit confusing. "Following an ethnographic and grounded theory approach data collection and analysis did not present independent successive but alternating phases." It's also confusing that you have some games used in the first approach, and then a subset in the second, and why it was a subset. I find the methods descriptions and rationales confusing and glossed over too much.

We added more detail to clarify the grounded theory approach and on the methodological approach of the ethnographic study.

R1: I am glad you explained the games. Should that be moved up earlier in the methods section? Could there be a chart of games and which methods you used with which?

We added a table to show which games were included in which studies. (We kept the section where it is because we felt it was important to first introduce how they are being studied.)

R1: Recommendations are great to see. I would include some with valuing the professions of game design, development, and community management more readily, with a plan of how to fund these as a professional, rather than through student work or interns.

We believe these recommendations are adequately captured in T3, T7, T10. 

R1: Can they get funding for a vendor? 

We added elaboration to recommendations of funding.

R1: Also, I know this is an area of expertise for some of you, but it is problematic when it says "gamified" throughout the paper-- these are games (CSGs) not gamified content. To me, they should be considered full-fledged games from the outset.

Here we use this term to mean making a game based on real science, rather than creating gamified content. We agree that CSGs are, for the most part, a gameful approach rather than a gamified one, but there is no better term for saying “turning a scientific task into a game” than “gamifying.”

R1: I like the table for recommendations. Could you separate it out as recommendations for development team, scientific community, educational community, etc?

For purposes of scope we have chosen to limit our recommendations to the development team only. As noted at the top of Section 5, we refer to prior literature for recommendations to other communities and stakeholders. 

R2: A more thorough description of the projects, with an explanation of the type of gaming employed (leader boards, collaborative, first-person, exploration etc.), and perhaps an investigation on how these types correlate to the opinions of stakeholders - this could be a whole follow-up paper however!

We added more detail to the games studied, ensuring that our descriptions of the games studied included the type of gaming employed and the discipline of focus. Unfortunately, as R2 says, correlations with the opinions of stakeholders would be an entire follow-up paper. 

R2: Perhaps a section on the disciplines involved in each project, how they relate to the school subjects, and if that correlates with any of the opinions of the students and/or teachers involved.

We added detail to ensure that each project’s discipline is clear. Unfortunately, there is insufficient data to make claims about correlations between game disciplines and how they relate to school subjects.

R2: There are now some very strong examples of citizen science being incorporated into commercial gaming, instead of the other way around (see MMOS, Project Discover on EVE online https://www.eveonline.com/discovery, and Borderlands 3 https://www.pcgamer.com/how-eve-online-and-borderlands-3-merge-citizen-science-and-minigames/) - where the game design and creation very much takes centre stage. This probably should at least be discussed at some point.

We added discussion on CSGs as being incorporated in commercial games.

Notably, we also added a section on data availability to ensure that our manuscript complies with the PLOS data policy.

We thank the reviewers for their time and hope that these revisions are satisfactory.

---

## [Editor Report · Decision Letter 1]

24 Apr 2023

Practical Recommendations from a Multi-Perspective Needs and Challenges Assessment of Citizen Science Games

PONE-D-22-30825R1

Dear Dr. Miller,

We’re pleased to inform you that your manuscript has been judged scientifically suitable for publication and will be formally accepted for publication once it meets all outstanding technical requirements.

Kind regards,

Peter Edwards

Academic Editor

PLOS ONE
---

## [Editor Report · Acceptance letter]

28 Apr 2023

PONE-D-22-30825R1 

Practical Recommendations from a Multi-Perspective Needs and Challenges Assessment of Citizen Science Games 

Dear Dr. Miller:

I'm pleased to inform you that your manuscript has been deemed suitable for publication in PLOS ONE. Congratulations! Your manuscript is now with our production department. 

Kind regards, 

on behalf of

Dr. Peter Edwards 

Academic Editor

PLOS ONE